

# Correlation of wind waves and sea level variations on the coast of the seasonally ice-covered Gulf of Finland

Milla M. Johansson, Jan-Victor Björkqvist, Jani Särkkä, Ulpu Leijala, and Kimmo K. Kahma

Finnish Meteorological Institute, P.O. Box 503, FI-00101, Helsinki, Finland

**Correspondence:** Milla Johansson (milla.johansson@fmi.fi)

**Abstract.** Both sea level variations and wind-generated waves affect coastal flooding risks. The correlation of these two phenomena complicates the estimates of their joint effect on the exceedance levels for the continuous water mass. In the northern Baltic Sea the seasonal occurrence of sea ice further influences the situation. We analysed this correlation with 28 years (1992–2019) of sea level data, and four years (2016–2019) of wave buoy measurements from a coastal location outside the City of Helsinki, Gulf of Finland. The wave observations were complemented by 28 years of simulations with a parametric wave model. The sea levels and waves at this location show strongest positive correlation ($\tau = 0.5$) for southwesterly winds, while for northeasterly winds the correlation is negative (-0.3). The results were qualitatively similar when only the open water period was considered, or when the ice season was included either with zero wave heights or hypothetical no-ice wave heights. We calculated the observed probability distribution of the sum of the sea level and the highest individual wave crest from the simultaneous time series. Compared to this, a probability distribution of the sum calculated by assuming that the two variables are independent underestimates the total water levels corresponding to one hour per 100 years by 0.1—1.2 m. We tested three Archimedean copulas, of which the Gumbel copula best accounted for the mutual dependence between the two variables.

## 1 Introduction

Urbanized and heavily populated coastal regions around the world face concrete consequences of sea level rise and climate change. To ensure safe and effective coastal protection and city planning in the future, accurate and location targeted estimates of coastal flood probabilities are in demand. The ultimate height to which the sea water rises in a coastal flood is determined by both the sea level variations – so called "still water level", including storm surges and longer-term variations – as well as wind-generated waves on top of that. Thus, when estimating coastal flooding risks, these both need to be taken into account.

On the Baltic Sea coasts the flooding risks have been extensively studied using long, high-quality tide gauge records, which on many locations date back to the 19th century (e.g. Johansson et al. (2001), Eelsalu et al. (2014), Soomere et al. (2015), Kulikov and Medvedev (2017); see also Hünicke et al. (2015) for an overview of sea level and wind wave studies in the



Baltic Sea). These data have enabled detailed analyses of sea level extremes and probability distributions useful for practical applications in coastal planning (e.g. Kahma et al., 2014).

To account for the additional effect of wind waves, Leijala et al. (2018) complemented the still-water-level-based flooding risk estimates with wave run-up on a steep shore. They estimated the probability distributions of both phenomena separately
and, assuming the two variables independent, calculated the probability distribution of their sum at two coastal locations close to Helsinki, Gulf of Finland. Even if the assumption of independence might not strictly hold, high sea levels and high waves do not necessarily co-occur, either. Hanson and Larson (2008) drew the conclusion that on the southern Swedish coast high run-up levels (sum of still water level and wave run-up) are typically caused by high waves in combination with modest still water levels.

The actual dependence of waves and sea level variations is complex. While short-term sea level variations and wind-generated waves are driven by related factors (namely wind and air pressure variations), strong winds can either fill or empty small sub-basins while simultaneously generating high waves. Also, the slowly changing total water volume of a semi-enclosed basin does not depend on the short term wind speed.

To account for the correlation between sea level variations and wind waves when estimating the exceedance frequencies of
their sum, a bivariate (two-dimensional) probability distribution is needed. Copulas are a flexible tool to study this geophysical problem, since they allow different types of marginal distributions of the individual variables, unlike e.g. multivariate Weibull or normal distributions. They have therefore been use to study storm surges in the German Bight (Wahl et al., 2012), the joint probability of water levels and waves at the Adriatic coast (Masina et al., 2015), and extreme coastal wave overtopping (Chini and Stansby, 2012). Kudryavtseva et al. (2020) used copulas to evaluate the likelihood of joint occurrence of high water levels
and wave heights along the whole Baltic Sea coast, showing that the probability of co-occurrence varies regionally, being highest in the Bothnian Sea, but also moderate in the Gulf of Finland.

The Gulf of Finland (GoF, Fig. 1) is a sub-basin of the Baltic Sea. The Baltic Sea is a semi-enclosed marginal sea connected to the North Sea and the North-Atlantic Ocean only through the narrow and shallow Danish Straits. The straits prevent rapid changes in the total water volume of the Baltic Sea, which therefore behaves like a closed basin on a sub-weekly time scale,
with the sea level varying internally. The internal variations amount to several tens of centimetres, and are mainly driven by wind and air pressure, but also by the consequent internal oscillations (seiches). These variations in the GoF are controlled by the local geometry of this narrow west-to-east oriented gulf, and can therefore behave differently depending on e.g. the wind direction (Leppäranta and Myrberg, 2009). The astronomical tides are small, with amplitudes less than 15 cm (e.g. Särkkä et al., 2017).

On time scales longer than a week the Baltic Sea water volume variations are a result of the water exchange that is low-pass filtered by the Danish Straits. This water level component is significant, with local sea level variations being up to 1.3 m (Leppäranta and Myrberg, 2009), while precipitation, evaporation and river runoff play a minor role. On even longer time scales, the Baltic Sea is affected by the global mean sea level rise, which is counterbalanced to varying extent by the local post-glacial land uplift (e.g. Pellikka et al., 2018; Vestøl et al., 2019).





The wind-generated waves in the GoF are influenced by the fetch, the seabed, and narrowness of the gulf (Kahma and Pettersson, 1994). The measured significant wave height has exceeded 5 m both during easterly and westerly winds (Pettersson et al., 2013; Pettersson and Jönsson, 2005), but the significant wave height close to Helsinki is expected to be less than half of the open-sea value, since waves are effectively attenuated by the coastal archipelago (Björkqvist et al., 2019).

5 Finally, both the sea level variations and the waves are affected by the seasonal ice season, which in the Baltic Sea is 5–7 months (Leppäranta and Myrberg, 2009). While the presence of ice somewhat attenuates sea level variability – mainly by blocking the piling-up effect of wind on the sea surface (Lisitzin, 1957) – it completely blocks waves if the ice concentration is high enough. The maximum ice coverage ranges from 12.5 to 100 % of the surface area of the sea (Leppäranta and Myrberg, 2009); on normal and severe ice winters the entire GoF gets ice-covered, while on mild winters only the eastern part of the gulf 10 freezes.

In this paper, we explore the correlation of sea level and wind waves near Helsinki on the northern coast of the Gulf of Finland, with an aim to determine its effect on the exceedance frequencies of their sum. The study is motivated by the high relevance of flood risk estimates for the City of Helsinki, as well as the recently obtained 4-year time series of coastal wave buoy data. These data, together with the multi-decadal time series of tide gauge observations, provide an excellent data set for 15 such case study. We aim at answering the following research questions:

1. To what extent are these two variables correlated, and does it depend on season, ice conditions or wind direction?

2. What is the observation-based probability distribution of the sum of the still water level and wave height? How much does it differ from a distribution formed assuming that the two variables are independent?

3. Can a copula-based bivariate distribution account for the correlation of the two variables, and more accurately describe 20 the probability of the sum?

This paper is structured as follows. In Sect. 2, we describe the observation data sets used, the parametric wave model, and the methods used in calculating the uni- and bivariate probability distributions. The results are presented in Sect. 3, followed by discussion in Sect. 4. We finally summarize our conclusions in Sect. 5.

## 2   Data and methods

25 ### 2.1   Sea level measurements

Automated tide gauges have been operational on the Finnish coast since the late 19[th] century, at Helsinki since 1904 (see e.g. Johansson et al. (2001) for a more detailed description of data, measurement techniques and quality). We used instant hourly sea levels measured in 1992–2019 at the Helsinki Kaivopuisto tide gauge (Fig. 1), currently operated by the Finnish Meteorological Institute (FMI). Missing values were patched by linear interpolation from the measurements at two other nearby 30 tide gauges on the Finnish coast (Hanko and Hamina). There were 477 interpolated hourly values in 1992–2019, less than 0.2

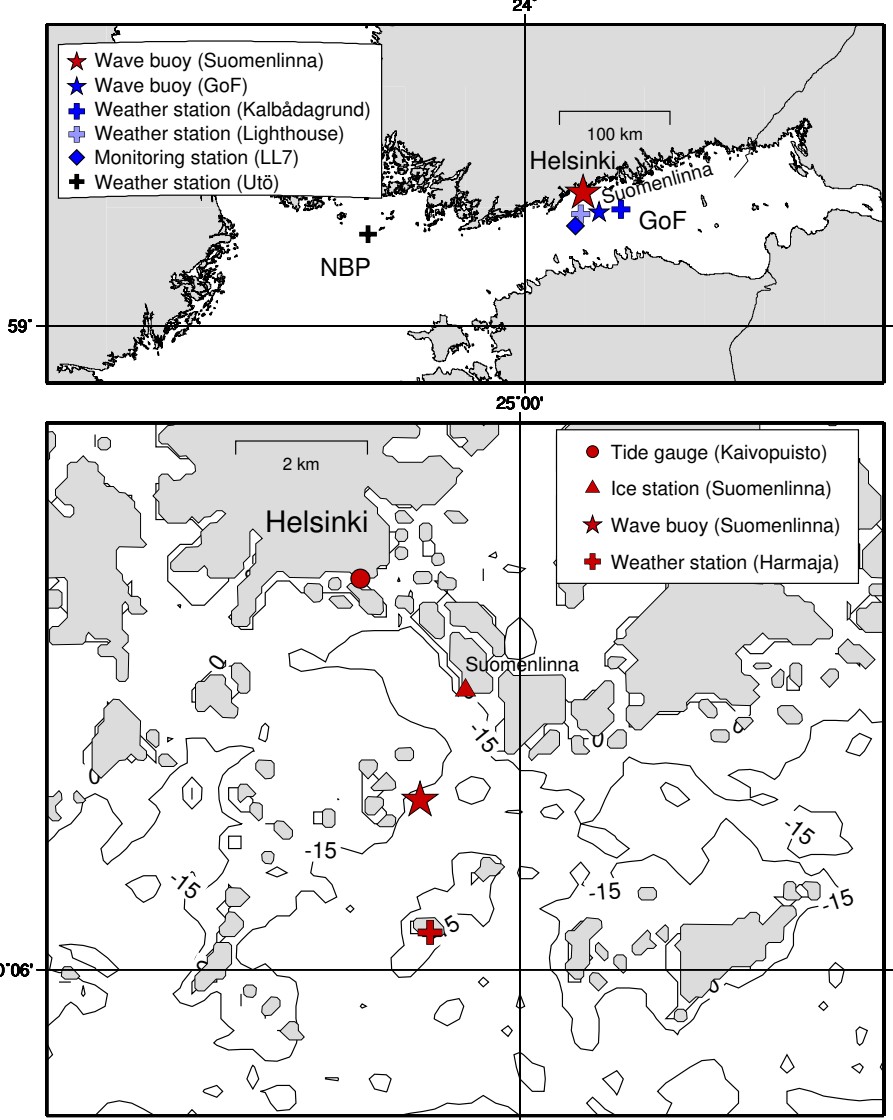

**Figure 1.** On top is a larger view of the study area covering the Northern Baltic Proper (NBP) and Gulf of Finland (GoF). The coastal area off Helsinki around Suomenlinna is shown on the bottom. The Suomenlinna ice station represents the ice conditions for about a 1 km radius.

% of the data. All sea levels in this study are given in relation to the N2000 height system, a Finnish realization of the common European height system (Saaranen et al., 2009).

The average sea level in 1992–2019 at Helsinki was +0.21 m, variations ranging from -0.73 to +1.69 m (Table 1). The linear trend in the time series was -2 cm in 28 years – the land uplift and sea level rise practically canceling each other. Thus, we can

5 safely ignore the non-stationarity introduced by the long-term trend.



## 2.2 Wind and temperature measurements

FMI's automatic weather stations provided wind speed, wind direction, and air temperature measurements (Fig. 1). From 1992 onwards these stations covered three areas of the Baltic Sea: i) Northern Baltic Proper (NBP) covered by Utö weather station (measuring height 19 metres above mean sea level), ii) Gulf of Finland (GoF) covered by Kalbådagrund and the Helsinki Lighthouse (31 and 33 metres above mean sea level), and iii) the coastal area near Suomenlinna, Helsinki covered by Harmaja (18 metres above mean sea level). The Helsinki Lighthouse weather station was only used for 2019 to cover a Kalbådagrund data gap.

We calculated hourly wind and air temperature values by either interpolating the older 3-hour data (up until mid to late 90s) or by calculating hourly averages of later denser measurements. All gaps shorter than six hours were interpolated. Continuous data blocks that were shorter than 12 hours were removed. Sea surface temperature (SST) data were only available from Harmaja starting end of June 1995. For other times and locations we estimated the SST using a mean yearly cycle from the GoF monitoring station LL7 (Fig. 1). The mean is defined from 434 CTD soundings from R/V *Aranda* between 1992 and 2017.

## 2.3 Observed wave heights

FMI's operational GoF wave buoy is located in the centre of the basin (62 m water depth), while the coastal wave buoy is anchored at a depth of 22 m outside of Suomenlinna, Helsinki (Fig. 1). While coastal wave measurements have been made in several places along the Finnish coast, this is the only permanent coastal wave buoy. Yearly measurements at these locations started in 2000 (GoF) and 2016 (Suomenlinna). In the Baltic Sea wave buoys are removed before the ice season, since freezing damages the sensors of the Datawell Directional Waveriders. The GoF buoy's measurement periods were 01 Jan – 18 Jan 2016, 12 Mar 2016 – 13 Jan 2017, 18 Apr 2017 – 29 Jan 2018, 14 Apr 2018 – 14 Jan 2019, and 11 Mar – 31 Dec 2019. For the Suomenlinna wave buoy they were 27 Apr – 29 Nov 2016, 20 Apr – 15 Dec 2017, 03 May – 26 Nov 2018, and 18 Apr – 10 Dec 2019.

The wave buoys determined the significant wave height from 27 minute vertical displacement time series as $H_s = 4\sqrt{m_0}$, where $m_0$ is the zeroth moment of the variance density spectrum (see Datawell, 2019). We picked every other value to get a data set with hourly time resolution. Of these values, the highest significant wave heights during 2016–2019 were 4.12 m (GoF) and 1.47 m (Suomenlinna); the mean values were 0.81 m and 0.33 m.

## 2.4 Simulated wave heights

We complemented the coastal wave measurements at Suomenlinna with 28 years (1992–2019) of simulations produced by a parametric wave model. The model uses fetch-limited wave growth relations (Kahma and Calkoen, 1992), while also accounting for the wind duration and the atmospheric stratification. A description of the wave evolution scheme is presented in Appendix A.

First we simulated the wave conditions at the GoF wave buoy, which were needed as boundary conditions for the coastal wave hindcast. The GoF simulation accounted for the locally generated waves in the middle of the gulf and waves propagating



from the eastern part of the basin and the Northern Baltic Proper – also simulated using the same model and the local wind. We then determined the fraction of the GoF wave energy that arrived to Suomenlinna (henceforth "long waves"). This long wave energy was quantified as the difference between the observed wave energy (for 2016) and the modelled local wave energy at Suomenlinna. The attenuation – as a function of the wind direction – was determined using a linear regression between

the simulated wave energy at the GoF and the estimated long wave energy at Suomenlinna. The final time series of coastal significant wave height was determined from the sum of the simulated local wave energy at Suomenlinna and the attenuated GoF wave energy.

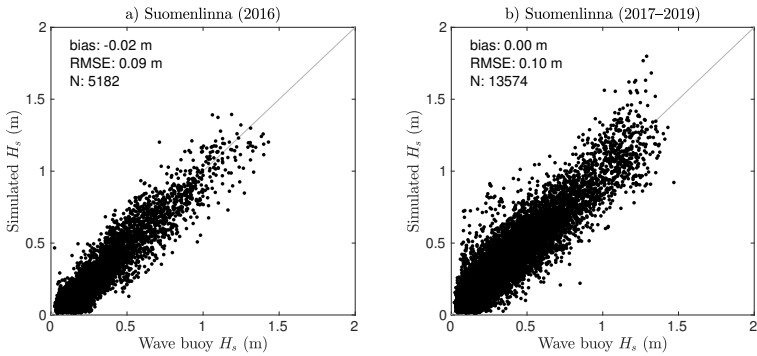

**Figure 2.** The significant wave height simulated by the parametric wave model compared with observations from the Suomenlinna wave buoys. The 2016 data used for model calibration are shown separately (a).

The simulated significant wave heights were validated using wave buoy data from 2016–2019. At GoF the -0.02 m bias and 0.28 m RMSE (N=25293) were slightly worse than for state-of-the-art numerical wave models (e.g. Tuomi et al., 2011;

Björkqvist et al., 2018). Nonetheless, the accuracy was sufficient to serve as boundary conditions for the coastal simulations at Suomenlinna, which had a 0.00 m bias and 0.10 m RMSE (N=13574) (Fig. 2). For Suomenlinna the validation statistics compare well with those from high-resolution numerical wave model simulations (Björkqvist et al., 2020), but the parametric model showed a slight tendency to overestimate the highest significant wave heights (Fig. 2 b). This is natural, since a linear regression cannot account for the increased wave–bottom interaction for the higher and longer waves during stronger winds

that were not captured in the calibration period. We can, however, conclude that the quality of the simulated wave height time series is adequate for the purpose of this study.

## 2.5 Ice conditions and wave statistics types

Data originating from FMI's ice charts described the ice conditions at the southern point of the Suomenlinna islands (Fig. 1). The data represent an area with a radius of roughly 1 km and is therefore fairly representative of the wave buoy location. Of the

statistics available for the entire study time (1992–2019) we used the start and end dates of the permanent ice-cover. During this time – defined as our ice period – the ice concentration is typically high (over 80 %), making it reasonable to set the significant





wave height to zero. The length of our ice period varied from zero (winters 2007–2008, 2017–2018, and 2018–2019) to 139 days (winter 2002–2003).

Tuomi et al. (2011) defined four different wave statistics for water bodies with a seasonal ice cover. Using this classification, we compiled the following data sets (Table 1):

– Data set F (ice-free time): simulated $H_s$ for the open water season only.

 – Data set I (ice-time-included): simulated $H_s$ set to 0 for the ice period.

 – Data set N (hypothetical no-ice): full simulated $H_s$ data set. As the wave model does not account for ice, this represents a hypothetical situation with no ice present.

 – Data set M (measurement statistics): wave observations only (this time is a true subset of the ice-free period because of
measurement gaps).

We divided the 28-year data sets (F, I, and N) into seven consecutive 4-year periods, from 1992–1995 to 2016–2019. These 4-year time series were, where applicable, used to study temporal variability.

**Table 1.** The sea level (still water level, $z_{still}$) and significant wave height ($H_s$) data sets used in this study.

| Parameter | Ice statistics | Symbol | Period | Amount of data | Mean (m) | Min / max (m) | Std (m) |
|---|---|---|---|---|---|---|---|
| Sea level | | $z_{still}$ | 1992–2019 | 245 448 | 0.21 | -0.73 / 1.69 | 0.24 |
| Observed $H_s$ | M | $H_{s,m}$ | 2016–2019 | 21 546 | 0.33 | 0.02 / 1.47 | 0.25 |
| Simulated $H_s$ | F | $H_{s,f}$ | 1992–2019 | 191 220 | 0.34 | 0.01 / 2.25 | 0.27 |
| | I | $H_{s,i}$ | 1992–2019 | 228 727 | 0.29 | 0.00 / 2.25 | 0.28 |
| | N | $H_{s,n}$ | 1992–2019 | 228 727 | 0.35 | 0.01 / 2.25 | 0.27 |

## 2.6  Probability distribution of the total water level

We define the *total water level*, $z_{max}$, as the highest level to which the continuous water mass reaches as a combined effect
of the still water level, $z_{still}$, and the wave action. Neglecting site specific coastal effects, we took the wave action to be the highest individual wave crest, $\eta_{max}$. At Suomenlinna the highest single wave crest during 30 minutes is approximately 92% of the significant wave height (Björkqvist et al., 2019), which we rounded up to $\eta_{max} = H_s$ for hourly values. Assuming a constant water level during each hour, the maximum water level elevation is

$$z_{max} = z_{still} + \eta_{max} = z_{still} + H_s. \tag{1}$$

The probability distributions of $z_{max}$ were then calculated directly from the time series obtained from the observed and simulated $z_{still}$ and $H_s$ data.




If simultaneous time series of $z_{still}$ and $H_s$ are not available, the cumulative distribution function (CDF) of $z_{max}$ needs to be calculated based on the respective univariate distributions. If the two variables are assumed to be independent (for dependent variables, see Sect. 2.7), the CDF of their sum is the convolution

$$F_z(z_{max}) = \int\limits_{\mathbb{R}} f_x(\xi) F_y(z_{max} - \xi)\, \mathrm{d}\xi \qquad (2)$$

where $f_x$ denotes the probability density function (PDF) of $z_{still}$, and $F_y$ denotes the CDF of $H_s$ (see Leijala et al. (2018) for more details).

## 2.7   Copula-based probability distributions

The Sklar's theorem (Nelsen, 2006) states that if $H$ is a bivariate CDF with marginal CDFs $F_1$ and $F_2$, then a copula C exists such that

$H(x_1, x_2) = C(F_1(x_1), F_2(x_2); \theta).$                                        (3)

Thus, the joint distribution $H$ can be expressed as a function of the marginal distributions $F_1$ and $F_2$ and the copula parameter $\theta$, which describes the dependency of the variables $x_1$ and $x_2$. If $x_1$ and $x_2$ are independent, the copula is simply the product $F_1(x_1)F_2(x_2)$. An extensive introduction to copula methods is provided by Genest and Favre (2007). A benefit of the method is that the marginal distributions can be of any form.

We calculated $\theta$ for three copulas from the Archimedean family: Clayton, Frank and Gumbel, which have commonly been used for hydrological analyses (e.g. Wahl et al., 2012; Kudryavtseva et al., 2020). We determined $\theta$ from the observed $z_{still}$ and $H_s$ data (2016–2019) with the inversion of the Kendall's $\tau$ estimator (see e.g. Wahl et al. (2012) for the relationship between $\theta$ and $\tau$). By using the bivariate distributions consisting of each of the three copulas and the marginal distributions (Sect. 2.8), we simulated $10^8$ random pairs ($z_{still}$, $H_s$) and calculated the CCDFs (complementary cumulative distribution functions) of

their sum. The calculations were performed with the R "copula" package.

## 2.8   Marginal distributions and extrapolation

Our 28 years of hourly data allow for a direct estimate of the frequency of exceedance down to $4 \cdot 10^{-6}$, and we therefore had to extrapolate the distribution to cover rarer values. Extremes are typically evaluated using some extreme value distribution (Eelsalu et al., 2014; Särkkä et al., 2017; Kulikov and Medvedev, 2017), but this approach is not applicable to extrapolate

hourly data needed for this study.

     The probability distribution of the observed sea levels on the Baltic Sea coasts is generally slightly asymmetric: the high-end tail is thicker than the low-end tail (e.g. Johansson et al., 2001; Kulikov and Medvedev, 2017). The significant wave heights in the archipelago, again, show a clear tendency to saturate (Leijala et al., 2018), and are therefore not modelled well simply by using some commonly used distribution type, such as the Weibull distribution (e.g. Battjes, 1972; Mathisen and Bitner-

Gregersen, 1990). We chose to extend the distributions to extreme values by extrapolating with two-parameter exponential





functions fitted to the tails of the CDFs and CCDFs (probability levels below 0.005), following e.g Leijala et al. (2018). This method was applied consistently to the sea level, significant wave height, and total water level data. The PDFs were obtained by numerically differentiating the CDFs.

## 3 Results

### 3.1 Correlation

The simultaneous hourly sea levels and significant wave heights show an apparent correlation (Fig. 3). The positive tail dependence – high sea levels co-occurring with high waves – is evident both in the observed and simulated data. The individual maxima, however, did not occur simultaneously. During the highest measured sea level (1.69 m) on 09 Jan 2005, the simulated wave height was a relatively modest 0.87 m. The highest observed wave height was 1.47 m on 07 Dec 2017, while the simulations gave 2.25 m on 23 Dec 2004. On those occasions, the sea levels were +0.73 m and +0.94 m, respectively: high but not extreme.

The variations in both parameters are largest in winter (DJF; Fig. 4) and smallest in summer (JJA), when the wave height did not exceed 1.5 m and the sea level was between -0.3 m and +1.0 m. The positive tail dependence can be seen in most of the seasonal data also.

The hypothetical no-ice data (type N) show situations with low sea level and wave heights ranging from 0 to 1 m which would have occurred during the ice-covered period, especially in DJF (Figs. 3 and 4). In MAM, also many high-sea-level and hypothetically high-wave situations occurred during the ice-covered time.

As a measure of the correlation, we used the Kendall's tau coefficient ($\tau$). It measures rank correlation: the similarity of the orderings of the data when ranked by each of the quantities. Thus, no assumption of a linear relationship between the variables is made. The sea level correlates stronger with the observed wave heights than the simulated wave heights from the same time period (the measurement seasons on 2016–2019) on all seasons except DJF (Table 2). The observed wave data in DJF are very sparse (Fig. 4a), practically only consisting of 15 days of observations in December 2017 and 10 days in December 2019, and thus the high correlation ($\tau = 0.3$) should not be considered representative. Likewise, the observations for MAM only represent late April and May, as the wave buoy was deployed earliest on 18 Apr and latest on 03 May. Of the different seasons, autumn (SON) shows the lowest correlation in all data sets.

The different ways of handling the ice time in the 28-year data sets (F, I, N) do not generally result in large differences in the correlation, the only exception being DJF where the data set I shows clearly higher correlation than data sets F or N. There is a tendency of lower sea levels during the ice-covered period (the blue points with $H_s = 0$ in Fig. 4a) which likely enhances the correlation.

The correlation clearly depends on wind direction measured at Harmaja (Fig. 5). The strongest positive correlation occurs with southwesterly winds (240–260°), while northeasterly winds (20–50°) result in zero or negative correlation. The behaviour is qualitatively similar in both the observed and simulated wave heights, and with all the data sets considered (F, I and N; results for data sets I and N shown in Appendix B).




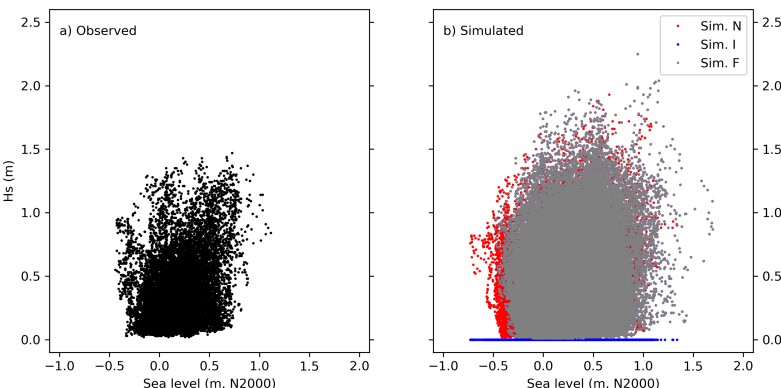

**Figure 3.** Scatter plot of significant wave heights vs. sea level: a) observed wave heights from 2016–2019 and b) simulated wave heights from 1992–2019, data sets F (gray), I (blue), and N (red). Note that every data point of the set F is also included in sets I and N.

**Table 2.** Kendall rank correlation coefficients ($\tau$) between the observed (Obs.) and simulated (Sim.) significant wave heights and sea levels on four seasons: winter (DJF), spring (MAM), summer (JJA) and autumn (SON). The data set "Sim. M" includes the simulated significant wave heights from the same time periods as the data set "Obs. M" contains observations, thus being a subset of "Sim. F" of the years 2016–2019.

| Data set | Years | All | DJF | MAM | JJA | SON |
|----------|-------|-----|-----|-----|-----|-----|
| Obs. M | 2016–2019 | 0.195 | 0.302 | 0.307 | 0.162 | 0.054 |
| Sim. M | 2016–2019 | 0.136 | 0.324 | 0.116 | 0.149 | 0.011 |
| Sim. F | 1992–2019 | 0.160 | 0.110 | 0.133 | 0.172 | 0.071 |
| Sim. I | 1992–2019 | 0.162 | 0.223 | 0.129 | 0.172 | 0.071 |
| Sim. N | 1992–2019 | 0.152 | 0.140 | 0.113 | 0.172 | 0.071 |

We analysed the correlation at different time scales by calculating the cross spectrum of the sea level variations and observed wave heights (Fig. 6). The spectra were calculated separately for each year 2016–2019, and Pearson correlations ($r$) for the frequency bins were determined by normalizing the co-spectra with the square roots of the individual power density spectra. Both the significant wave height and the sea level spectra showed a peak at the one day mark. Compared to the waves, the sea level variations weaker for sub-day times scales relative to multi-week time scales (Fig. 6a). The correlation was strongest ($r \approx 0.4$) for time scales between one and three days, with variation shorter than one day being completely de-correlated. There was a weak correlation ($r = 0.2$) for time scales longer than a week, but multi-month variations were not captured by the ca. 8 month time series after the applied Blackman-Harris window tapering and smoothing of the spectra.

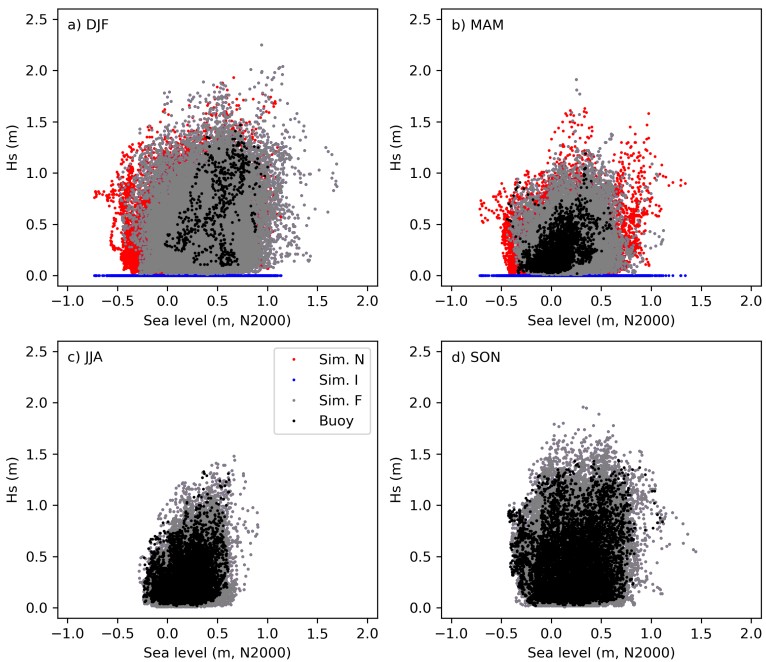

**Figure 4.** Scatter plots of a) winter, b) spring, c) summer, and d) autumn events. Black: observed wave heights from 2016–2019. Other colors: simulated wave heights from 1992–2019, ice types F (gray), I (blue), and N (red).

## 3.2 Probability distributions

The CCDF of sea level from 2016–2019 shows lower high tail than the CCDF from the entire period 1992–2019 (Fig. 7a). The 2016–2019 distribution is also on the lower limit of the ensemble of the seven different four-year distributions. The period 2016–2019 had less high sea levels than most of the four-year periods, having also the lowest four-year maximum (1.11 m) of all these periods. The distribution from 2004–2007 has a distinct tail with sea levels exceeding 1.45 m, while the maxima of all the other periods are below this. All these high values resulted from the exceptional storm Gudrun on 09 Jan 2005, when the sea level at Helsinki stayed above 1.45 m for 13 hours from 01 to 13 UTC.

The shape of the CCDF obtained from the observed wave heights differs from those obtained from the simulated wave heights (Fig. 7b), the probability of exceedance decreasing more rapidly with increasing wave height in the observed CCDF than in the simulated ones. This is in accordance with the parametric wave model validation (Fig. 2d) which showed an overestimation of highest significant wave heights.





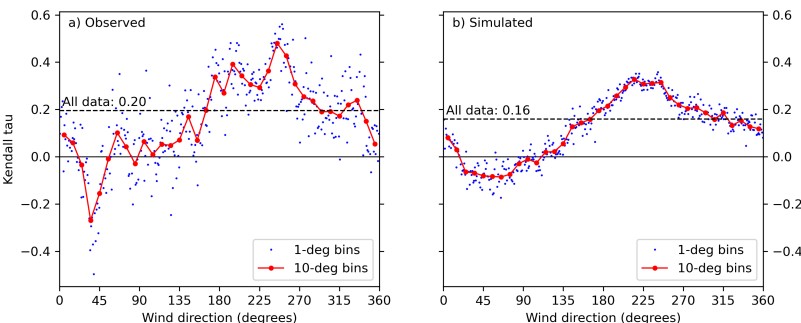

**Figure 5.** The Kendall correlation coefficients ($\tau$) between significant wave height and sea level for different wind directions. The blue dots show correlation in 1-degree direction bins, and the red dots in 10-degree bins. The significant wave heights from buoy observations 2016–2019 (left) and simulated data set F 1992–2019 (right) are shown. The direction given is the direction from where the wind blows, observed at Harmaja weather station. The correlation coefficients for all data ($\tau$ = 0.20 and 0.16) are shown as horizontal lines.

The CCDFs of $z_{max}$ calculated by summing the observed sea levels and observed or simulated wave heights according to Eq. (1), and extrapolating with exponential functions, show similar behaviour at the high tail (Fig. 7c): the distribution based on observed wave heights has lower tail than any of those based on simulated wave heights. The distributions of the sum of $z_{still}$ and $H_s$ obtained by assuming the two variables independent, according to Eq. (2) (Fig. 7d) give lower probabilities for high sea

levels than the distributions calculated from the sum of the observed sea levels and the observed/simulated wave heights (Fig. 7c). At the probability level $1.14 \cdot 10^{-6}$, which corresponds to one hour per 100 years, the difference in the 4-year distributions ranges from 0.14 m to 1.24 m, and at the level $1.14*10^{-7}$ (one hour per 1000 years) from 0.16 m to 1.52 m. The distributions based on the observed buoy data 2016–2019 differ by 0.36 m (1/100) and 0.42 m (1/1000), and the distribution based on the 28-year simulation by 0.69 m (1/100) and 0.84 m (1/1000).

The CCDFs based on data sets F, I and N show only minor differences (Fig. 7 for F, others in Appendix B). This indicates that the hypothetical wave and sea level behaviour during the ice-covered time does not differ from that during the open water period. The amount of ice days, however, is small compared to the length of the study period: 28 days (1.9 %) in 2016–2019 and 1615 days (16 %) in 1992–2019. Thus, it is expected that they do not significantly alter the results.

### 3.3   Copula-based distributions

Table 3 shows the parameters of different Archimedean copulas fitted to the sea levels and significant wave heights. In Fig. 8, CCDFs obtained from these copulas and the marginal distributions are compared with the observed CCDF and the CCDF obtained by assuming the two variables independent. In accordance with the above results, the observed CCDF shows higher probabilities of high total water levels than the independent case (here represented by the independence copula, results for which are identical to the results obtained by the convolution method, Eq. 2). Frank and Clayton copulas behave only marginally

better than the independent case, apparently not being able to reproduce the effect of correlation on the CCDF. The Gumbel



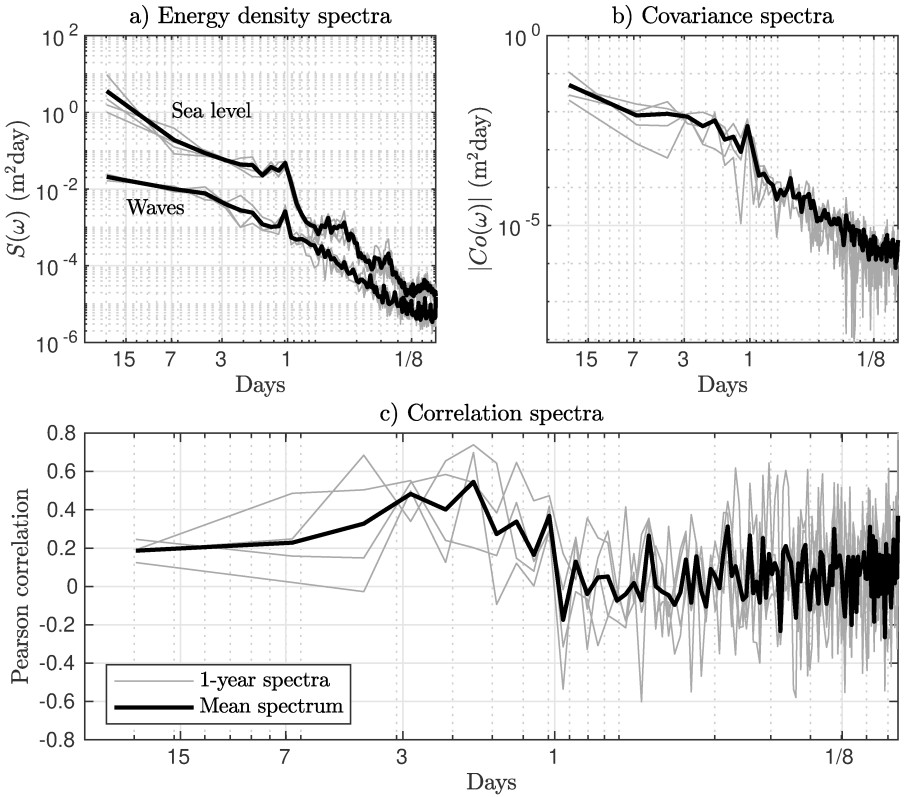

**Figure 6.** Power density spectra (a), covariance spectra (b), and correlation spectra (c) of the measured significant wave height at Suomenlinna (2016–2019) and the coinciding sea level variations.

copula is able to reproduce the observed CCDF on the more common cases, down to probability level $10^{-2}$, but even it fails to reproduce the downward bending of the tail on lower probabilities. The Kolmogorov-Smirnov test statistic confirms this, being lowest for the Gumbel copula -based distribution.

To further illustrate the differences in bivariate distributions based on different copulas, we simulated an ensemble of 21 546 samples from each, these being of the same size as the observed data set (Fig. 9). It is visually apparent that Gumbel copula is the only one which reproduces the positive tail dependence of the observed scatter: the slight bending of the scatter cloud into a tail on the upper right corner. The Frank and Clayton copulas do not show such a tail; instead, the Clayton copula shows a negative tail in the lower left corner which is stronger than in the observations. In the independent case, no tail is present as expected.

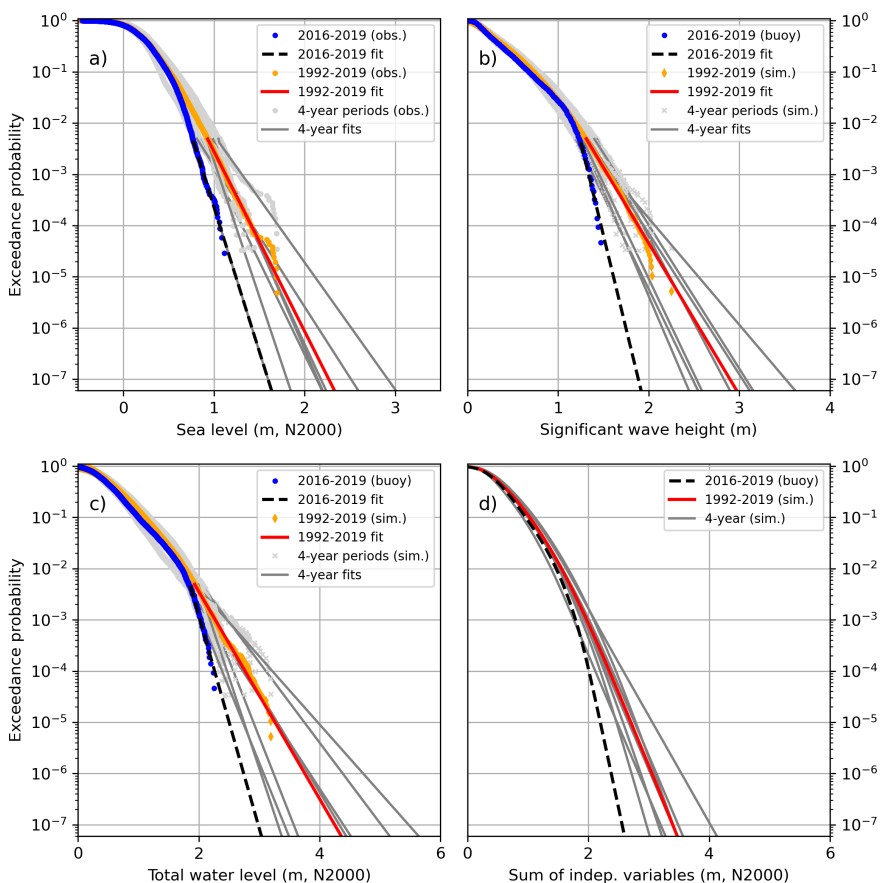

**Figure 7.** The CCDFs of a) observed hourly sea levels, b) observed/simulated significant wave heights, c) total water level as their sum and d) total water level calculated as a sum of two independent variables. The distributions shown are those for the data set F (ice-free season). The distributions were extrapolated with exponential functions below the probability level 0.005. The seven consecutive 4-year periods shown separately for observed sea levels and simulated waves are 1992–1995 to 2016–2019.

**Table 3.** Fitted parameters of three different types of Archimedean copulas for sea levels and significant wave heights from buoy observations (Obs.) and simulated data (Sim.) for 2016–2019 with different ice statistics.

| Hs type | Ice type | $\tau$ | N | Gumbel $\theta$ | Frank $\theta$ | Clayton $\theta$ |
|---------|----------|--------|-------|-----------------|----------------|------------------|
| Obs. | M | 0.195 | 21546 | 1.242 | 1.812 | 0.485 |
| Sim. | F | 0.148 | 31269 | 1.174 | 1.359 | 0.348 |
| Sim. | I | 0.148 | 31789 | 1.174 | 1.358 | 0.348 |
| Sim. | N | 0.150 | 31789 | 1.176 | 1.372 | 0.352 |

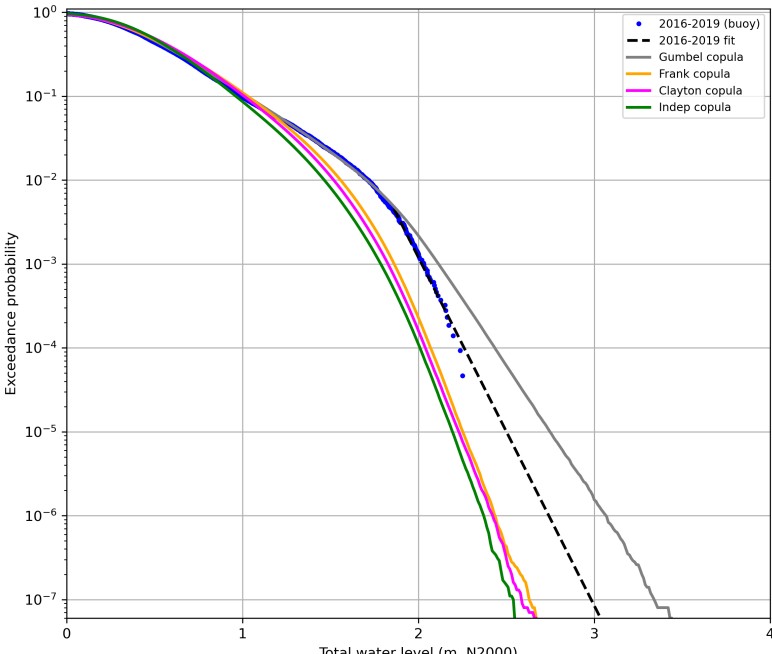

**Figure 8.** The observation-based CCDF of total water level in 2016–2019, as well as CCDFs calculated by taking $10^8$ random samples based on the three Archimedean copulas and the independence copula combined with the observation-based marginal distributions.

## 4 Discussion

As the wave conditions in the archipelago differ markedly from those on the open sea, a multi-year time series of wave data from a coastal location such as Suomenlinna provides an excellent material for studies. At the time of this study, measurements from four ice-free seasons were available, but we expect the time series to continue. As more data accumulate, it will provide
5   more comprehensive knowledge on the local wave behaviour.

We compensated for the shortness of the wave buoy time series by using 28 years of simulated wave data. The lightweight parametric wave model is considerably faster to run than high-resolution numerical wave models. Nevertheless, the bias and RMSE of the simulations were comparable. The discrepancy of the model results and observations in the highest significant wave heights, however, naturally affects any estimates of the probabilities of extremely high total water levels.

10   One of the advantages of a wave model is also its ability to simulate the wave conditions during ice-covered periods, or periods when there is a risk of freezing and the wave buoy cannot be kept on sea. This extends the time series and provides an opportunity to analyse "what would happen if there was no ice?" Such question is relevant when we consider the conditions


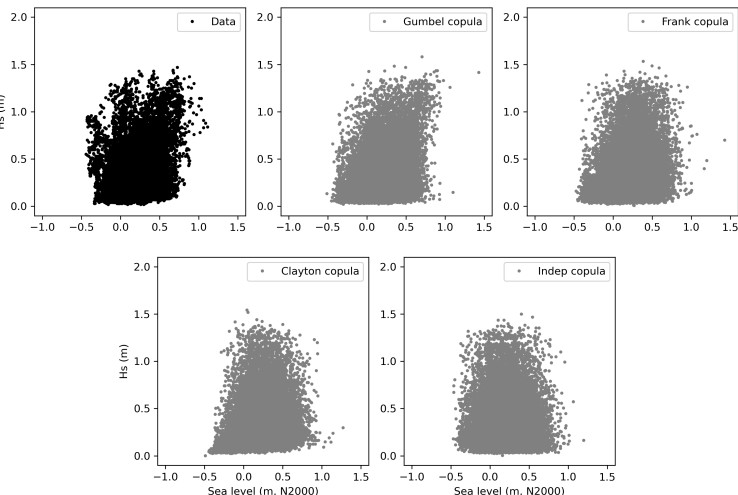

**Figure 9.** Scatter plots of the observed sea level ($z_{still}$) and significant wave height ($H_s$) in 2016–2019 (top left), and scatters of 21 546 random points (equalling the count of observations) sampled from the bivariate distributions based on three Archimedean copulas and the independence copula.

in the GoF in a warmer future climate, for instance. Our results did not indicate that the wave or sea level conditions, or the correlation of these, would markedly differ from those occurring during the ice-free period in the present climate.

The wave conditions vary significantly among 4-year periods (Fig. 7). E.g. the 4-year maximum significant wave height varied from 1.74 m to 2.25 m; a scatter of more than 0.5 m. The sea levels show a comparable variation, the 4-year maxima
5  ranging from 1.11 to 1.69 m. Especially, it is noteworthy that the observational period on which this study was based (2016–2019), was relatively peaceful when considering sea level variations or waves in the GoF. The 4-year maximum sea level (1.11 m) was lowest among the seven consecutive 4-year sets, while the maximum simulated significant wave height (1.89 m) ranked fourth. To obtain reliable estimates for probabilities of extreme sea levels on the Finnish coast, at least 30 years of data are generally used (e.g. Pellikka et al., 2018; Kahma et al., 2014). Our results obtained from 4-year time series can only be
10  considered a methodological analysis on the applicability of different methods for obtaining exceedance probabilities of high total water levels. They do not provide any flooding risk estimates for practical applications.

Additionally, considering flooding on the coast, the relevant parameter is the wave run-up. While our total water level (Eq. 1) represents the elevation of the highest wave crest on the open sea, it is not applicable on the waves meeting the coastline. On a steep coast, a more appropriate estimate for the total water level would be $z_{still} + 2 \cdot H_s$ due to wave reflection (Leijala
15  et al., 2018), while on a shallow coast the eventual run-up on the coastline is less than $z_{still} + H_s$.



The behaviour of the correlation with respect to wind direction (Fig. 5) is readily explained by the geography of the GoF. Southwesterly winds push water into the gulf and simultaneously raise high waves, leading to positive correlation. Northeasterly winds, on the other hand, push water out from the GoF while still raising waves, which shows up as a negative correlation. It is possible that when the correlation is calculated from the entire data set, these two opposing situations with different mech-

anisms somewhat cancel each other and reduce the overall correlation. To more completely describe the effect of correlation on the distribution of the total water level, and thus the flooding risks, one option might be to use three variables instead of two: still water level, significant wave height and wind direction. We leave this as an option for future studies.

The sea level processes in the GoF act on different time scales, from the sub-daily storm surges to sub-weekly Baltic Sea internal variations, and weekly and longer-term changes in the total water volume. These can be used to interpret the differences

in the correlation for different time scales (Fig. 6). The sea level and wave responses to wind or pressure forcing have time delays of some hours: it takes time for the water to flow into the GoF, and wind-generated waves to develop. Different response times to the atmospheric forcing could explain the low correlation on sub-daily time scales.

From one to seven days, the Baltic Sea mainly responds to wind and air pressure forcing like a closed basin. The response time of such variations, e.g. the transport of water between the GoF and the Baltic Proper in a seiche oscillation, is usually of

the order of one day. In such time scale, the wind-generated waves have time to develop, and these phenomena co-occur.

The sea level variations in 7–90 days usually involve changes in the Baltic Sea water volume. These are related to longer-term (up to 60 days; Särkkä et al., 2017) prevailing wind conditions, which control the in- or outflow of water in the Danish Straits. These should not have a direct relationship with the local waves. However, the longer-term weather conditions correlate with short-term weather phenomena too: e.g. prevailing westerly winds are related to more low-pressure systems with strong

winds travelling over the Baltic Sea. This shows up as a weaker, but still existing correlation.

Our results demonstrated that the assumption of independence of sea level and significant wave height leads to an underestimation of the total water levels corresponding to certain probabilities of exceedance. The probability level of $1.14 \cdot 10^{-6}$, for instance, corresponds to one hour in 100 years, which is a relevant level for flooding risk considerations. The total water levels corresponding to this probability were underestimated by 0.7 m in the 28-year data set, and 0.1–1.2 m in the 4-year data sets

(the difference of the distributions in Figs. 7c and 7d). With lower probabilities of exceedance, the underestimation generally increases. This underlines the need to take the correlation into account when estimating probabilities of water level extremes on the coast. It turned out that a suitable copula function – such as the Gumbel copula – is able to incorporate the effect of the correlation on the CCDF to some extent. Further studies are needed to ascertain the applicability of the copula approach, and to find the optimal copula type.

**5   Conclusions**

Our main conclusions, related to the research questions presented in Sect. 1, are as follows.

The sea level variations and significant wave heights show a positive correlation in general. The observed sea levels correlate both with the observed and the simulated wave heights. This correlation shows a dependence on wind direction: southwesterly





winds lead to strongest positive correlation, while northeasterly winds lead to zero or negative correlation. Ice conditions did not have an apparent effect on the correlation: including hypothetical no-ice wave heights during the ice season did not markedly alter the correlation, and neither did the setting of the wave height to zero during the ice-covered time.

The 4-year probability distributions of the total water level, calculated as a sum of the observed sea level and observed/simulated
wave heights, give 0.1–1.2 m higher values for extremely high sea levels (one hour in 100 years) than the distributions calculated by assuming the two variables independent. This underlines the importance of accounting for the dependence between the variables when calculating the probabilities of high total water levels e.g. for flooding risk estimates.

The probability distribution of the sum based on the copula approach is closer to the observed distribution than the distribution based on the independence assumption. Of the three Archimedean copulas studied, the Gumbel copula proved the
most suitable for our data. However, more studies are needed before we can say whether the copula approach is suitable for calculating the probability distributions, and which copula would be optimal.

*Code and data availability.* The wave, wind and sea-level observations are available through FMI's open data portal. The code for the parametric wave model will be made available in a public repository upon the publication of the final paper.

## Appendix A: Model scheme for wave evolution

The core idea of the wave model is based on known universal properties of different dimensionless parameters, namely the energy, peak frequency, phase speed, fetch, and time:

$$\tilde{\varepsilon} = g^2\varepsilon/U^4 \tag{A1}$$
$$\tilde{f}_p = f_pU/g \tag{A2}$$
$$\tilde{c}_p = c_p/U \tag{A3}$$
$$\tilde{X} = gX/U^2 \tag{A4}$$
$$\tilde{t} = gt/U. \tag{A5}$$

The wave growth in the model is quantified using the fetch limited wave growth relations of Kahma and Calkoen (1992):

$$\tilde{\varepsilon} = p_2\tilde{X}^{p_1} \tag{A6}$$
$$\tilde{f}_p = q_2\tilde{X}^{q_1}. \tag{A7}$$

The coefficients $p_1, p_2, q_1, q_2$ for the wave growth relations depend on the atmospheric stratification as presented by the authors.



Instead of assuming that the waves are fetch limited at each time step, the model simulates a more realistic wave growth and decay by also accounting for the limitation of wind duration through the following propagation scheme:

1. Calculate the dimensionless fetch, $\tilde{X}_{\varepsilon^{i-1}}$, based on the energy from the *previous* time step, $\varepsilon^{i-1}$, and the *current* wind speed, $U^i$.

2. Calculate a (dimensionless) duration based on the wind, $U^i$, and Eq. A7 as:

$$\tilde{t}_0 = \int\limits_{0}^{\tilde{X}_{\varepsilon^{i-1}}} \left(\frac{1}{2}\tilde{c}_p\right)^{-1} \mathrm{d}\tilde{X} = \int\limits_{0}^{\tilde{X}_{\varepsilon^{i-1}}} 4\pi\tilde{f}_p \, \mathrm{d}\tilde{X} = \int\limits_{0}^{\tilde{X}_{\varepsilon^{i-1}}} 4\pi q_2 \tilde{X}^{q_1} \, \mathrm{d}\tilde{X} = \frac{4\pi q_2}{q_1+1}\tilde{X}_{\varepsilon^{i-1}}^{(q_1+1)}. \tag{A8}$$

3. Determine an enhanced dimensionless time step as $\Delta\tilde{t} + \tilde{t}_0$, where $\Delta t$ is the fixed time step of the model (e.g. 1 h).

4. As an inverse to step 2, calculate a new fetch $\tilde{X}_t$ based on the enhanced duration $\Delta\tilde{t} + \tilde{t}_0$, and Eq. A7 as:

$$\tilde{X}_t = \left(\frac{(\tilde{t} + \tilde{t}_0)(q_1+1)}{4\pi q_2}\right)^{1/(q_1+1)}. \tag{A9}$$

5. Calculate the minimum dimensionless fetch for a fully developed sea (using the wind speed at 10 metre height) as:

$$\frac{U}{c_p} = \frac{2\pi f_p U}{g} = 2\pi\tilde{f}_p = 0.82 \Longleftrightarrow \tilde{X}_{FD} = \left(\frac{0.82}{2\pi q_2}\right)^{1/q_1}. \tag{A10}$$

6. Set the relevant fetch to $\tilde{X}^i = \min\{\tilde{X}_t, \tilde{X}_{FD}, \tilde{X}\}$, where $\tilde{X}$ is the physical dimensionless fetch.

7. Calculate the new energy, $\varepsilon^i$, and peak frequency, $f_p^i$, using $\tilde{X}^i$, $U^i$, and Equations A6 and A7.

## Appendix B: Supplementary figures

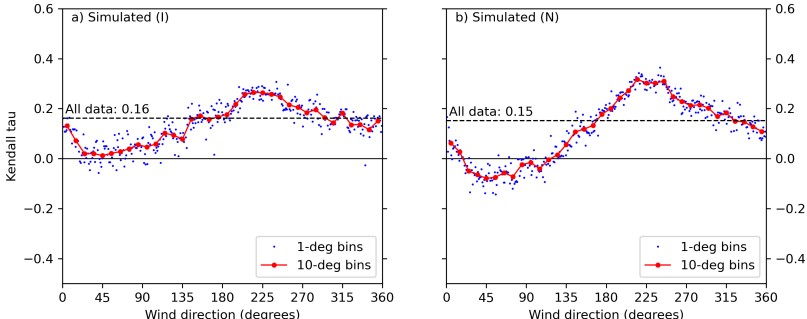

**Figure B1.** The Kendall correlation coefficients ($\tau$) between significant wave height and sea level for different wind directions. The blue dots show correlation in 1-degree direction bins, and the red dots in 10-degree bins. The significant wave heights from simulated data sets I (left) and N (right) 1992–2019 are shown. The direction given is the direction from where the wind blows, observed at Harmaja weather station. The correlation coefficients for all data ($\tau = 0.16$ and $0.15$) are shown as horizontal lines.

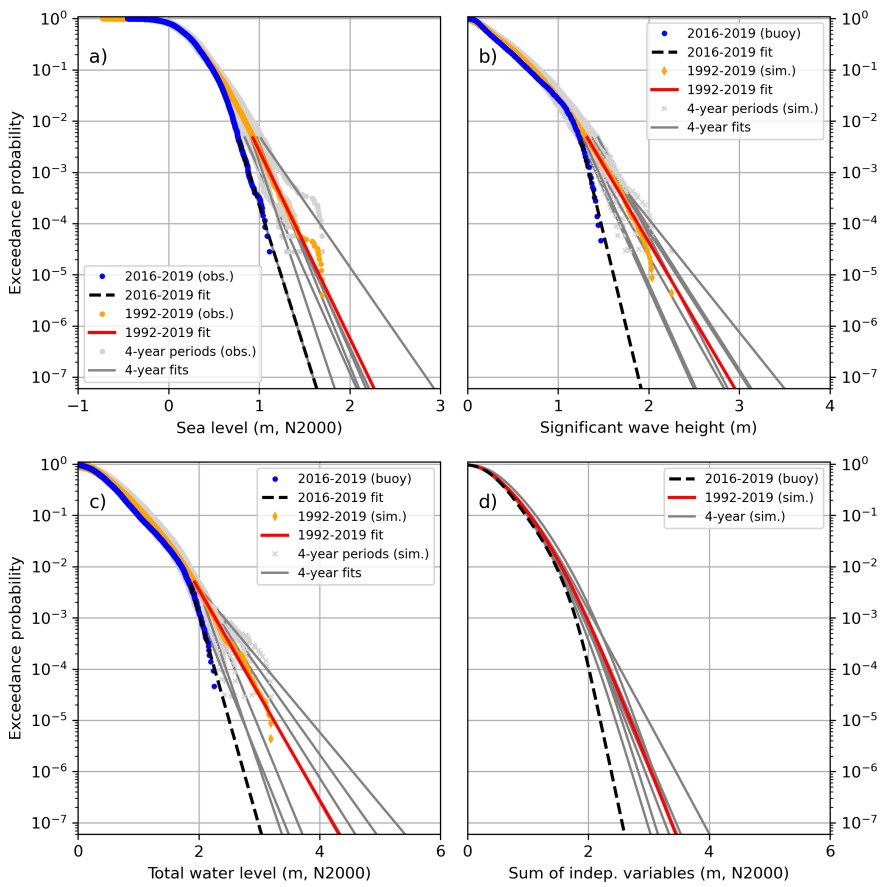

**Figure B2.** The CCDFs of a) observed hourly sea levels, b) observed/simulated significant wave heights, c) total water level as their sum and d) total water level calculated as a sum of two independent variables. The distributions shown are those for the data set N (hypothetical no-ice). The distributions were extrapolated with exponential functions below the probability level 0.005. The seven consecutive 4-year periods shown separately for observed sea levels and simulated waves are 1992–1995 to 2016–2019.


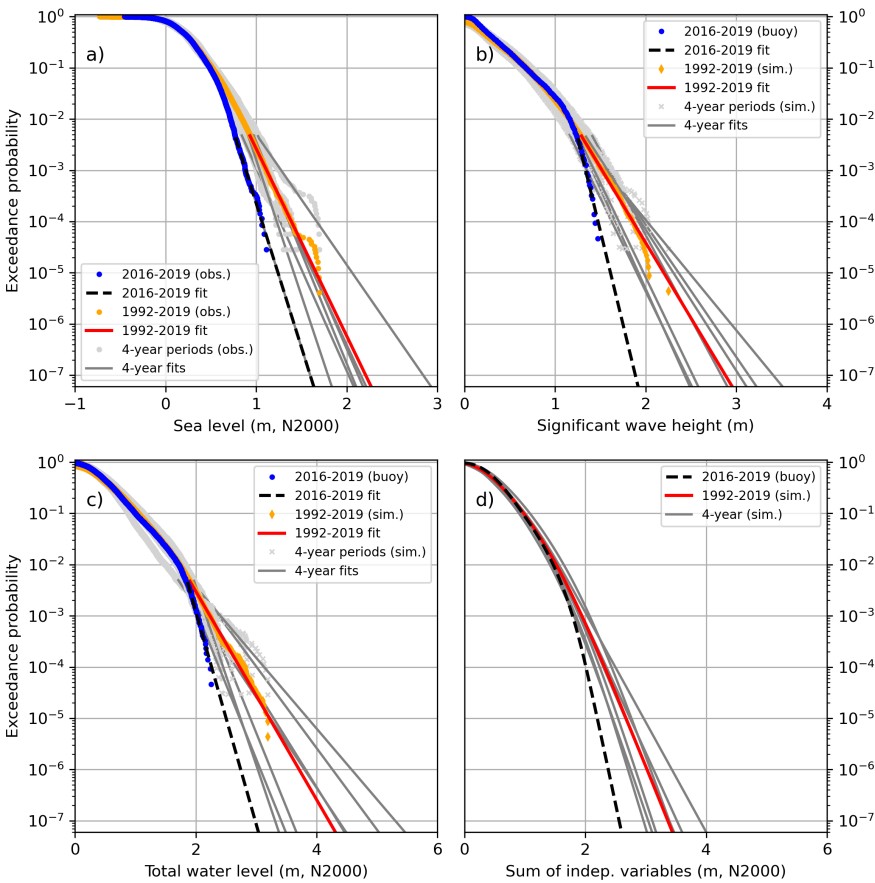

**Figure B3.** The CCDFs of a) observed hourly sea levels, b) observed/simulated significant wave heights, c) total water level as their sum and d) total water level calculated as a sum of two independent variables. The distributions shown are those for the data set I (ice time included). The distributions were extrapolated with exponential functions below the probability level 0.005. The seven consecutive 4-year periods shown separately for observed sea levels and simulated waves are 1992–1995 to 2016–2019.

*Author contributions.* MJ calculated the correlations, probability distributions and copulas and wrote most of the paper; JVB programmed the coastal implementation of the parametric wave model, did the model runs and spectral calculations and contributed to writing; JS started the work and wrote the first draft; UL contributed to the calculation of the probability distributions; KK originally developed the parametric wave model and proposed the idea of this study.

*Competing interests.* No competing interests are present.



*Acknowledgements.* This work was partly funded by the State Nuclear Waste Management Fund in Finland through the EXWE project (Extreme weather and nuclear power plants) of the SAFIR2018 programme (The Finnish Nuclear Power Plant Safety Research Programme 2015–2018; http://safir2018.vtt.fi). The work done by Pekka Alenius in providing the water temperature data for the LL7 monitoring site is greatly acknowledged.





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
