# Peer review of "Correlation of wind waves and sea level variations on the coast of the seasonally ice-covered Gulf of Finland"

_Natural Hazards and Earth System Sciences, 2021_

## Author Response (AR1)

**Responses to Reviewer #1**

We thank Reviewer #1 for the constructive comments which helped us to improve the manuscript. Below, the reviewer comments are given in *italics*, and our responses in plain text.

*This manuscript uses observations of sea level, wind and waves to infer their relationships in the Gulf of Finland. The authors compute rank correlations and bivariate distributions (with and without dependence) between sea level and significant wave heights.*

*I have some concerns regarding the methods applied and the interpretation of the results, that I outline below, with some suggestions for improvement. Some of these suggestions may require big changes in the manuscript, though. Some moderate to minor comments follow.*

*First, when fitting the marginal distributions, in section 2.8, exponential functions are used for sea level and wave height. The exponential function has one parameter to fit, so I assume that the 2 parameters here refer to the two marginal CCDFs (one parameter each). I am not convinced that exponential functions are the most appropriate for extreme value analyses, which are the values the authors intend to extrapolate. Statistically, they may indicate better performance when using tests as AIC or BIC simply because the number of parameters to be fitted is smaller (one vs 2 or 3 in GEV, etc). But this does not mean that the fitting is better. In fact, extreme distributions do converge to these families of distributions (provided they fulfil the hypotheses). This choice is based on the results by Leijala (2018) which are in turn based on Särkkä (2017). The only reason provided there to use the exponential function was the number of parameters, but I do not think this is justified enough. Also, to me, one major issue here is why are the authors fitting the entire dataset to a distribution if the focus is on extremes. I think it would be better to select only those values considered as extremes in either one or another variable. That would make the rest of the computations easier too.*

We added the formula of the two-parameter exponential function to the manuscript: F = exp(-λ(x-x0)).

Our purpose in this study was to study the correlation between simultaneous sea level and wave height in general, not just the correlation of the extreme events (which may differ). Thus, we chose to use the entire data set of hourly values. The frequency distribution of these is not an extreme value distribution, and clearly does not qualify for fitting GEV, for instance. It is not well defined which form the distribution of hourly sea levels follows, and speculating with the best-fitting distribution is not in the core focus of this study (although it is an interesting topic for further studies). Thus, we chose a simple two-parameter function which fits the data.

So that our results would better highlight the fact that we are not focusing on extremes only, we chose to show the probability levels 1/10 and 1/100 h yr$^{-1}$ and leave the 1/1000 h yr$^{-1}$ values out.

*In the results on the correlations (section 3.1) I disagree with the statement that Hs and sea level are correlated. The correlation is not apparent from figure 3, since values of Hs exceeding 1 m may occur with nearly any value of sea level. The analysis provided later looking at correlations as a function of wind direction is more meaningful.*

This is true, the correlation in the entire data set is not apparent. We removed the mention of this and changed the text in 3.1 to: "The simultaneous hourly sea levels and significant wave heights (Fig. 3) indicate some positive tail dependence -- high sea levels co-occurring with high waves -- both in the observed and simulated data. Though high waves also occur with moderate or even low sea levels, and high sea levels are not necessarily accompanied by high waves either."

*Actually, Figure 5 is very illustrative. However, I do not understand the results. When wind blows from the SW then there exists correlation between Hs and sea level at the buoy in Suomenlinna; but this buoy has some islets in its SW so it should be on their shadow. Am I missing something? or perhaps I am misinterpreting the angles. In any case it would be useful to specify the convention used for the directions in the text.*

It is true that there are some small islands shadowing the wave buoy location in the south-west. There are also larger islands to the southeast. These surrounding islands certainly provide shelter, but there still a significant amount of energy that are propagated around them, especially the longer waves. This is evident form e.g. Figure 4 in Björkqvist et al. (2020).

As for the correlation, it is more driven by the sea-level dynamics than the waves. Strong winds from both southwest and east can generate waves, but only westerly winds will simultaneously drive up the sea level. The easterly winds actually empties the Gulf of Finland, hence the positive correlation for southwesterly winds and negative correlation for easterly winds.

References: Björkqvist, J.-V., Vähä-Piikkiö, O., Alari, V., Kuznetsova, A., and Tuomi, L., 2020: WAM, SWAN and WAVEWATCH III in the Finnish archipelago – the effect of spectral performance on bulk wave parameters, J. Oper. Oceanogr., 13, 55–70, DOI: 10.1080/1755876X.2019.1633236

We added the convention to the figure caption: "The direction given is the direction from where the wind blows (nautical convention), …"

*Still in section 3.1, I do not think that computing the correlations between spectra is the way to calculate how the correlation changes for different timescales in the time series. Cross-spectra provides that information partly. The alternative is to filter the original records using band-pass filters of the desired frequencies and then correlate each of them. Also, when correlations are stated, it is necessary to provide confidence intervals. I very much doubt that 0.2 is a significant correlation, for example. I am not sure that this part of the section is meaningful, at least not in its present form.*

We decided to remove this part of Section 3.1 from the manuscript.

*In section 3.2, when using all hourly values within the time series, any of the copulas seem to fit the observations satisfactorily, as shown in figure 7. Given the variability of the correlations with wind (and thus wave) direction, it would make sense to restrict the analyses to those periods of time when the two variables do show a coherent behaviour. This would surely improve the estimates.*

We repeated the copula analysis separately for the data corresponding to southwesterly-wind situations, and northeasterly-wind situations. It turns out that the southwesterly-wind results are qualitatively similar, although with stronger correlation, than the results for the entire data set. The very low or non-existent correlation in the northeasterly-wind data leads to the copula method being rather useless.

We added to the text: "We tested the effect of wind direction on the copula results by doing the copula analysis separately for two subsets of the data: situations with southwesterly wind (160°--340°, Fig. 7b) and northeasterly wind (340° --160°, not shown). The results for the southwesterly-wind subset are qualitatively similar to those including all data: copulas are divided into two groups. The stronger correlation (tau = 0.269) results in somewhat higher probabilities of high total water levels in the distributions. In the northeasterly-wind case, on the other hand, the non-existent correlation (tau = -0.012) leads to the copula analysis being somewhat pointless, as all the copulas end up close to the independent case."

*In summary, I think the authors are working with ta data set for which it is worth exploring the dependences. In my opinion this should be done differently though. First, the authors should consider using only extreme for both variables. This reduces the number of data but the relationship is clearer. Second, extreme should be fitted with a suitable distribution for extremes. Third, events of either sea level or waves caused by different wind directions are very likely to belong to different families of distribution, since they probably arise from different atmospheric perturbations (e.g. they travel in distinct directions). This implies that the data should be treated and analysed separately. Copula functions should be also fitted for every subset in terms of direction and using the corresponding rank correlation. There are statistical tests to select the best copula fit, in case they show similar performance when compared to the observations. The comparison to the independent case is useful but cannot be taken as realistic if the Kendall correlation is high. Finally, I would suggest to remove the correlations of the spectra.*

The proposed analysis of extreme values is indeed interesting, but the aim of this study was to investigate the two processes more broadly. Also see our response above.

We repeated the copula analysis for two different data sets for different wind directions, see response above.

We added a goodness-of-fit analysis for the copulas by calculating the Cramér-von Mises statistics.

We removed the correlations of the spectra as suggested.

*Other comments:*

*-Page 2, lines 17-19: there are many others: Wahl et al (2015) (https://doi.org/10.1038/NCLIMATE2736) for rain and storm surges; Arns et al (2017) (https://doi.org/10.1038/srep40171) for surges and waves; Marcos et al (2019) (10.1029/2019GL082599) for surges and waves too but globally. And references therein...*

We added these references to the Introduction.

*- equation 1 in page 7: this is instantaneous water level at a water depth of around 20 m, where waves are measured/modelled. Sea level maxima are generally defined over periods longer than just a few seconds, so the exact meaning of z must be clearly specified to avoid misinterpretations.*

We clarified the text to read: "We defined the total water level, $z_{max}$, as the hourly maximum level to which the continuous water mass reaches as a combined effect of the still water level, $z_{still}$, and the

wave action. To do this, we first assumed that $z_{still}$ does not change during an hour. Neglecting site specific coastal effects, $z_{max}$ in such case is determined by the highest individual wave crest, $\eta_{max}$. (Note that wave crest denotes the height above $z_{still}$ and is thus half of the actual height of the highest wave.) At Suomenlinna the highest single wave crest during 30 minutes is approximately 92 % of the significant wave height (Björkqvist et al., 2019), which we rounded up to $\eta_{max} = H_s$ for hourly values. With these assumptions, the maximum water level elevation is…"

*-p. 8, l. 20: please, provide the full name of the library used*

We added the full name and proper references.

*-p.8, l. 22: units of the frequency are missing (I guess h^-1).*

We changed the units of all frequencies of sea level and significant wave heights to *h yr$^{-1}$*. The units of the total water level, which is effectively a hourly maximum, were given as events/year. We also clarified this in the text:

"The distributions of $z_{still}$ and $H_s$ both describe the actual duration of the phenomenon, and thus have units of h yr$^{-1}$. The distribution of $z_{max}$, on the other hand, has a unit of events/year, where "event" is defined as the instant hourly maximum total water level.

*-p.10, l.3-4: normalization will not impact the Pearson correlations*

This part of the manuscript was removed based on a previous comment.

*-p.10, l.5: weaker->weaken?*

This part of the manuscript was removed based on a previous comment.

*-p. 15, l.8-9: the bias of the model in extreme waves is not discussed enough. This is an important shortcoming of this work*

The overestimation of the highest values by the model is a shortcoming, but this is, nonetheless, not a study of the behaviour of extreme values. Our work studies the correlation of water level variations and surface waves more generally, and since the extreme values only account for a small fraction of the data, the main results are not dependent on the model performance for the highest wave heights. Indeed, Figure 5 shows that the correlation found in the observations is well reproduced using the model data. Also, the results from the Copulas are based on observations, and are therefore not tainted by possible model errors.

We have added a paragraph to the discussion that directly addresses this aspect of the model performance:

"The validation suggests that the highest wave heights are overestimated in the simulations. The attenuation of longer waves through bottom processes is indirectly accounted for in the model through the calibration against measurements. Nonetheless, this calibration might not hold for even harsher weather conditions and even longer waves. While the simulations offer a good tool for assessing the

general dependence between the wave height and water level variations, they should not be used as is for analyzing extreme values. If the wave simulations of the parametric model are to be used for extreme conditions, the wave-bottom interactions need to be accounted for in a more explicit manner, and such improved model needs to be re-validated."

*-p. 17,l.33: the observed sea levels correlate-> actually, this is true only for particular prevailing wind conditions.*

We changed the text to: "The sea level variations and significant wave heights show a positive correlation in general (tau = 0.20). The correlation depends on wind direction: southwesterly winds lead to strongest positive correlation (up to tau = 0.5), while northeasterly winds lead to zero or negative correlation."

*-p.18, l. 2-3: "including hypothetical no-ice wave heights during the ice season did not markedly alter the correlation,". This is the expectation, right? why would this change if the relationship wind and waves remain the same?*

We agree that this is not a surprising result, but also not immediately obvious because of two reasons:

1) The winds are somewhat stronger during the ice-time, which means that the hypothetical waves simulated for the ice time (but without including ice) might be higher than the waves during the ice-free time.

2) The sea level variations that are measured during the ice time are damped because of the ice, and are probably therefore smaller than during the ice-free time.

Points 1) and 2) together means that the correlation of the two variables during the hypothetical ice-free time is not trivially the same as for the ice-free time. The ice-time is short enough that this doesn't have a large impact, but we still feel it is worth to try to quantify the effect, especially since we expect that the seasonal ice-cover might change in the future.

**Responses to Reviewer #2**

We thank Reviewer #2 for the constructive comments which helped us to improve our manuscript. Below, the reviewer comments are given in *italics*, and our responses in plain text.

*This paper assesses the role of sea level and wind waves in generating total water level events in the Gulf of Finland, which is covered by ice during certain times of the year, making it a more challenging but also very interesting analysis. I think the content of the manuscript is novel and deserves publication with NHESS. While most of the analysis is technically sound and well-presented there are some aspects which I think require a bit more work before the paper can be recommended for publication. I summarize these below split into one major general comment and several (mostly minor) specific comments.*

*General comment:*

*I find the copula analysis part of the paper pretty weak. The authors decide to only use three copulas, without providing convincing arguments for that selection other than pointing to previous literature. In the past especially people using Matlab ended up focusing on the Archimedean copulas as those were implemented and easy to use. However, the development of the MvCAT copula toolbox by Sadegh et al. (2017) has made it much easier to draw from a larger set of copulas. In R there are also many more copulas readily available to use. More important than using a larger set of copulas would be to show whether or not the copulas that are used at the moment are actually capable of capturing the dependence structure of the observations. There are many different goodness-of-fit tests available. Without any such tests blindly applying a random set of copulas does not provide relevant insight. I strongly encourage the authors to invest a little bit more time in strengthening this part of the analysis, as I find it to be an important component (otherwise it could just be left out but that puts a hole into the analysis).*

We extended our analysis to include nine different copulas. We also calculated a goodness-of-fit test based on the Cramér-von Mises statistic.

*Specific comments*

*P1, l6 make clear in the abstract that Hs is used to represent waves*

We added "significant wave height" to the abstract.

*P1, l11 the one hour per 100 years sounds a bit strange; I understand the reasoning but isn't it just a 100-year event in the end?*

We changed the sentence to a more general form: "Compared to this, a probability distribution of the sum calculated by assuming that the two variables are independent underestimates the exceedance frequencies of high total water levels."

*P1, l18 It would be good to already mention here when defining total water levels that tides are negligible, not everyone will know that and wonder about the definition*

We added this to the text: "On the Baltic Sea coasts, the still water level variations mainly consist of storm surges and longer-term variations, the amplitude of tides being small."

*P1, l21 "in many locations"*

Corrected.

*P2, l25 why not say "decimeters"?*

Corrected.

*P3, l29 consider replacing "patched" by "inferred"*

We changed this to "estimated".

*Fig. 1 mention in caption what the contours are*

We added this to the caption: "The contours show the water depth in metres."

*P4, l5 is the trend also statistically insignificant?*

The trend is statistically significant due to large number of data points. Our intention was to say that its magnitude (-2 cm) is negligible compared to the amplitude of the variations. Statistical significance plays no role in this. We changed the paragraph to read:

"The average sea level in 1992–2019 at Helsinki was +0.21 m, variations ranging from -0.73 to +1.69 m (Table 1). The linear trend in the time series was -2 cm in 28 years – the land uplift and sea level rise practically canceling each other. These long-term phenomena thus have a negligible contribution to the sea level variations studied here."

*Sect. 2.2 mention somewhere early on why temperature is needed (it's not a typical variable one would use in most areas; recalling that it is used to determine ice-free periods would be helpful)*

We added the explanation that the temperatures were used as a forcing for the parametric wave model. The ice-free periods were determined from the ice charts.

*P6, l2 not sure if "long waves" is a good term here as it is basically reserved for long period waves (not travel distance)*

You are right, and we are actually referring to long period waves in the sense that they are longer than what can be generated by the local wind and the nearest fetch. This mix of different wave systems is

typical in the archipelago (Björkqvist et al. 2019). We now realize the ambiguity and will change the text in the manuscript to the following:

"We then determined the fraction of the GoF wave energy that arrived to Suomenlinna; these waves have longer periods than what can be generated by the local wind and the nearest fetch, and will henceforth be called "long waves"."

*P7, l16 I had a hard time following this definition, if Hs is the average of the 33% highest waves how is the highest single wave lower than that?*

We clarified the text to explain that wave crest (above the still water level) is only half of the wave height:

"Neglecting site specific coastal effects, $z_{max}$ in such case is determined by the highest individual wave crest, $\eta_{max}$. (Note that wave crest denotes the height above $z_{still}$, thus being half of the actual height of the highest wave.) At Suomenlinna the highest single wave crest during 30 minutes is approximately 92 % of the significant wave height (Björkqvist et al., 2019), …"

*Eq. 3: using theta which is special to the Archimedean (and one parameter) copulas makes that an alteration of Sklar's theorem*

This is true. We removed theta from the equation.

*P8, l19 first spell out and then introduce the abbreviation CCDFs*

Corrected.

*Fig. 3 when describing the results it would be helpful to recall that sim I has wave heights set to zero*

We added this to figure caption.

*P10, l5 "sea level variations are weaker"*

This part (analysis of the spectra) was removed from the manuscript based on a comment from Reviewer #1.

*Fig. 5 caption: the values for tau from the entire data are not in table 3 (the one for observations of 0.2 is included but just not rounded, but the 0.16 is not)*

The tau values (0.20 for observations 2016-2019, and 0.16 for simulations 1992-2019, set F) are given in Table 2. We added a reference to Table 2 in the figure caption.

*Fig. 7 & 8: make clear that it y-axis shows exceedance probability per hour (or translate values to exceedance probability per year)*

We translated the values in the y axes of the figures to "h yr$^{-1}$" or "events/year, corrected the text accordingly, and added an explanation in Section 2.6:

The distributions of $z_{still}$ and $H_s$ both describe the actual duration of the phenomenon, and thus have units of h yr$^{-1}$. The distribution of $z_{max}$, on the other hand, has a unit of events/year, where "event" is defined as the instant hourly maximum total water level.

*Fig. 8 It would be good to show the convolution results from figure 7c as well for direct comparison (only the one for 1992 to 2019); also see general comment above about testing which (or any) of the copulas are actually a good choice*

In Fig. 8, the convolution results from Fig. 7 (for the observations 2016-2019) are also shown. We clarified this in the caption: "The observation-based CCDF of total water level in 2016–2019 ("buoy" + "fit"), the CCDF calculated assuming the variables independent ("Sum of indep."), as well as CCDFs calculated by taking $10^8$ random samples based on the nine different copulas…"

We added a goodness-of-fit test for the copulas based on the Cramér-von Mises statistic.

*P15, l8 "is the highest"*

We changed this to "for the highest".

*P16, l6 "quite" is a better term then "peaceful"*

We changed this to "calm".

*P16, l6-8 that sentence wasn't clear to me please explain better what it's supposed to tell the reader and why it's relevant*

We tried to tell the reader that the 4-year period of observations seems to be on the calmer side when considering all possible 4-year periods of the study. This then reinforces that it is not representative in itself, and longer time series are needed for robust conclusions. We have modified this section by removing a lot of the numbers and only keeping the core message. We hope that this improves the readability. It now reads:

"The observational period of this study (2016--2019) was relatively calm: the 4-year maximum sea level (1.11 m) was lowest among the seven consecutive 4-year sets, while the maximum simulated significant wave height (1.89 m) ranked fourth. To obtain reliable estimates for probabilities of extreme sea levels on the Finnish coast, at least 30 years of data are generally used [...]"

*P17, l22-25 In the discussion further up the authors correctly point to the fact that the observed data is way too short to infer information for longer return periods (I would have commented on that if I hadn't seen that remark); why not focus (or at least add) results for a more reasonable return period, such as 10 years or so? At least we know the results would be more robust.*

We changed the manuscript so that we only show results for return periods of 10 and 100 years, and left 1000 years out.

---

## Author Response (AR2)

**Responses to Reviewer #1**

Below, the reviewer comments are given in *italics*, and our responses in blue plain text.

*I have gone through the revised version and, while I appreciate the changes made and the corresponding responses provided by the authors, I find this work is still confusing in some parts. My major concern in the first review was the choice of the probability function for the marginal distributions of sea level and waves, which has been defined as an exponential function. I argued that it may not be the best one for extremes. The authors have responded that the extreme analysis is not the purpose of this work, but in fact, they extrapolate their statistics to values of low probabilities, thus discussing extreme events. This is evident from (some examples):*

We have modified the manuscript to concentrate more on the part of the distribution which is not strongly affected by the extrapolation (generally from 100 down to 0.1 events/year). See more detailed responses below.

*- Second conclusion in section 5: "This underlines the importance of accounting for the dependence between the variables when calculating the probabilities of high total water levels e.g. for flooding risk estimates"*

We reworded this to read: "As the total water level values rarer than 0.1 events/year are also likely underestimated, the dependence between the variables should be accounted for when calculating the probabilities of high total water levels e.g. for flooding risk estimates."

*- page 19, line 1: exceedance frequency of 0.01 events/yr*

We replaced this with a discussion about less extreme values in the distribution, on frequency interval 100-0.1 events/year.

*- page 18, line 5: "Our results obtained from 4-year time series can only be considered a methodological analysis on the applicability of different methods for obtaining exceedance frequencies of high total water levels"*

We removed this sentence and instead say just: "Our results obtained from 4-year time series do not provide any flooding risk estimates for practical applications."

*- page 13, lines 5 to 10*

We replaced the results on extremes with results concerning the part of the distribution which is not so strongly dependent on the choice of the extrapolation function (from 100 down to 0.1 events/year).

*- section 3.3: discussions about the tails of the bivariate distributions*

This section discusses the differences in the copula-based distributions in general; e.g. the Cramér-von Mises statistics. We reworded the section to clarify that the results here are not relying on the extreme tails of the distributions.

*- Figures 6 and 7: extrapolation to very low probability values*

We changed the figure axes to remove the lowest extrapolations.

*- Etc.*

*Also, in page 9, l. 14-15: this sentence seems to be in contradiction with the manuscript central idea and with the conclusion stated in the last 2 lines of page 18.*

We changed the discussion. Instead of the extremes, we now discuss the underestimation of the total water levels in the frequency interval 100-0.1 events/year, which are more soundly based on the observed distributions and not dependent on the extrapolation function.

*So, after reading the manuscript again, my interpretation remains similar. The authors are fitting an exponential function to the entire time series, which provides a good fit because values around the mean dominate. But this does not mean that the same distribution can be used to infer the statistics of the tail of the distribution. Ideally, one should use a combination of two different distributions for different parts of the histogram. Or at least, perform two separate analyses for the mean and extreme regimes. An alternative, which is perhaps less demanding, is to discuss this point in the manuscript and make it explicit when describing the method.*

The reviewer's suggestion here seems to be in fact what we have done in the analysis. We used three separate functions for different parts of the distribution: for frequencies above 44 h/yr (the mean regime, 0.5-99.5 %), the empirical distribution function was used. The exponential functions were fitted to the frequencies below 44 h/yr (either CDF for the low tail of the distribution, or CCDF for the high tail.) We clarified the manuscript to point this out:

"We used the empirical frequency distributions for exceedance frequencies between the 0.5 and 99.5 percentiles, which correspond to a frequency of 44 h yr$^{-1}$ in the tail of the CDF/CCDF. We chose to extrapolate the high (low) tails of the distribution by fitting two-parameter exponential functions $F = e^{-\lambda.(x - x0)}$ to the CCDFs (CDFs) below this frequency, following e.g Leijala et al. (2018). This method was applied consistently to the sea level, significant wave height (only high tail), and total water level data. The PDFs were obtained by numerically differentiating the CDFs. As our main focus in this study is not on the extrapolated extreme ends of the distribution, we left more detailed considerations of the extrapolation function out and considered the simple exponential function sufficient for our purposes."

*In section 3.3, I understand that the authors have included a number of Copula functions to respond to another reviewer's request. However, some of them are clearly not suitable, so what is the purpose of showing the results that do not fit the data? In my opinion it would be enough to mention that 9 have been tested and only a subset of the best fits is shown.*

We agree that it would be possible to show only the best fitting copulas. However, on our opinion, it is better to show explicitly how the different copulas behave. This provides the reader an opportunity to see the difference between suitable and not suitable copulas, not just rely on our subjective assessment.